# Noise reduction and quantification of fiber orientations in greyscale images

**Maximilian Witte**[1,2], **Sören Jaspers**[2], **Horst Wenck**[2], **Michael Rübhausen**[1], **Frank Fischer**[2]*

**1** Center for Free-Electron Laser Science (CFEL), University of Hamburg, Hamburg, Germany, **2** Beiersdorf AG, Hamburg, Germany

* frank.fischer@beiersdorf.com

**Data Availability Statement:** All relevant data are within the manuscript and its Supporting Information files. All images are available from the figshare database (DOI: 10.6084/m9.figshare. 10529879).

## Abstract

Quantification of the angular orientation distribution of fibrous tissue structures in scientific images benefits from the Fourier image analysis to obtain quantitative information. Measurement uncertainties represent a major challenge and need to be considered by propagating them in order to determine an adaptive anisotropic Fourier filter. Our adaptive filter method (AF) is based on the maximum relative uncertainty $\delta_{cut}$ of the power spectrum as well as a weighted radial sum with weighting factor $\alpha$. We use a Monte-Carlo simulation to obtain realistic greyscale images that include defined variations in fiber thickness, length, and angular dispersion as well as variations in noise. From this simulation the best agreement between predefined and derived angular orientation distribution is found for evaluation parameters $\delta_{cut} = 2.1\%$ and $\alpha = 1.5$. The resulting cumulative orientation distribution was modeled by a sigmoid function to obtain the mean angle and the fiber dispersion. A comparison to a state-of-the-art band-pass method revealed that the AF method is more suitable for the application on greyscale fiber images, since the error of the fiber dispersion significantly decreased from $(33.9 \pm 26.5)\%$ to $(13.2 \pm 12.7)\%$. Both methods were found to accurately quantify the mean fiber orientation with an error of $(1.9 \pm 1.5)°$ and $(2.3 \pm 2.1)°$ in case of the AF and the band-pass method, respectively. We demonstrate that the AF method is able to accurately quantify the fiber orientation distribution in in vivo second-harmonic generation images of dermal collagen with a mean fiber orientation error of $(6.0 \pm 4.0)°$ and a dispersion error of $(9.3 \pm 12.1)\%$.

## Introduction

The evaluation of the angular distribution of structures in scientific greyscale images is of major importance for various applications like in the analysis of soft tissue fibers e.g. in [1–15], textiles [16, 17], electrospun scaffolds [18–20] or even reinforced concrete [21]. Knowing the angular distribution in fiber reinforced materials gives meaningful insights into their mechanical functionality [22]. For example, in case of biological tissue, the orientation distribution of collagen fibers can be directly inserted into biomechanical material models for finite element simulations [23, 24].

**Funding:** Beiersdorf AG provided support in the form of salaries for authors MW, SJ, HW and FF, but did not have any additional role in the study design, data collection and analysis, decision to publish, or preparation of the manuscript. The specific roles of these authors are articulated in the 'authors contribution' section.

**Competing interests:** We have the following interests: MW, SJ, HW and FF are employed by Beiersdorf AG. There are no patents, products in development or marketed products to declare. This does not alter our adherence to all the PLOS ONE policies on sharing data and materials, as detailed online in the guide for authors.

Fiber images from different image modalities such as scanning electron microscopy [25], histology [1], and laser scanning microscopy as e.g. in [2, 3, 7, 8, 10, 15, 22, 26–29] provide diverse image properties in terms of sharpness, contrast and fiber appearance. Thus, a variety of automated image processing techniques were developed including ellipsoidal fitting in the spatial domain [30], fiber tracking [31], the structure tensor method [10] and Fourier domain image processing [7, 9, 12, 32–34].

Looking into the Fourier method in more detail, it can be split into four major steps: image preprocessing, Fourier transform and filtering, calculation of the angular distribution and its quantification.

In the Fourier domain it is of key importance to reduce noise as it was shown that rotationally symmetric band-pass filtering significantly improve the accuracy of the method [6, 33, 34]. In mechanical experiments, (e.g. stretching of fiber reinforced material), fiber properties such as angular distribution and diameter get modified [2, 35]. Accordingly, isotropic as well as anisotropic signal responses in the Fourier domain has to be accounted for, which requires an adaptive anisotropic filtering. Quantification of the angular orientation distribution in terms of mean angle and fiber dispersion is commonly realized by fitting a semi-circular von-Mises distribution to the angular orientation distribution [6, 10, 12, 22, 30, 34, 36, 37].

Here, we investigate the Fourier method by exploiting objective approaches at any of the four mentioned steps. To approach the requirement of adaptive anisotropic filtering we introduce an image filter, which is based on the propagation of measurement uncertainties through the discrete Fourier transform. And finally, in terms of an improved quantification of the angular orientation distribution, we test a sigmoid function to model the cumulative orientation distribution.

To capture the performance of our method, we quantitatively compare it to a band-pass method, introduced by Morrill et. al [34], which was proven to provide an accurate quantification of the angular orientation distribution. Based on realistic Monte-Carlo simulated grey-scale fiber images we observe the evolution of accuracy with respect to fiber width, fiber aspect ratio, degree of alignment and image quality.

To validate the applicability of our method on real images, we quantify the mean fiber orientations and dispersions in vivo second-harmonic generation (SHG) images of collagen fibers of human skin [38].

## Material and methods

Any calculations were performed using MATLAB [39]. The *Image Processing Toolbox* [40] as well as the *Curve Fitting Toolbox* [41] were applied.

Note that in the following the term *uncertainty* is used for statistical measurement errors, whereas the term *error* is associated with the deviation of a value to its reference.

### Fourier transform and uncertainty propagation

Let $I(x, y)$ be an image with $x \in [0, X]$ and $y \in [0, Y]$. The discrete Fourier transform $I(x, y) \to \mathcal{F}[I(x, y)] = \hat{I}(u, v)$ is given by:

$$\hat{I}(u, v) = \sum_{y=0}^{Y-1} \sum_{x=0}^{X-1} I(x, y) \cdot e^{-2\pi i \left(\frac{x}{X}u + \frac{y}{Y}v\right)} \quad , \tag{1}$$

where the real and imaginary parts read as

$$\Re[\hat{I}(u, v)] = \sum_{y=0}^{Y-1} \sum_{x=0}^{X-1} I(x, y) \cos\left(2\pi i \left(\frac{x}{X}u + \frac{y}{Y}v\right)\right) \tag{2}$$

$$\mathfrak{I}[\hat{I}(u, v)] = -\sum_{y=0}^{Y-1}\sum_{x=0}^{X-1} I(x, y) \sin\left(2\pi\mathrm{i}\left(\frac{x}{X}u + \frac{y}{Y}v\right)\right) \tag{3}$$

The intensities of the angular distribution are calculated by evaluating the centered power spectrum of $I(x, y)$, which is defined as the squared magnitude of $\hat{I}$:

$$\mathcal{P}(u, v) = |\mathcal{F}[I(x, y)]|^2 = \mathfrak{R}[\hat{I}]^2 + \mathfrak{I}[\hat{I}]^2 \tag{4}$$

Any intensity image exhibits a specific intensity uncertainty $\Delta I(x, y)$, which is at least equal to the photon counting uncertainty $\Delta I(x, y) = \sqrt{I(x, y)}$ assuming Poissonian statistics [42]. The uncertainty of the real and imaginary part, $\Delta\mathfrak{R}$ and $\Delta\mathfrak{I}$ are given by propagating Eqs 2 and 3:

$$\Delta\mathfrak{R}[\hat{I}(u, v)] = \sqrt{\sum_{y=0}^{Y-1}\sum_{x=0}^{X-1}\Delta I^2(x, y) \cos^2\left(2\pi\mathrm{i}\left(\frac{x}{X}u + \frac{y}{Y}v\right)\right)} \tag{5}$$

$$\Delta\mathfrak{I}[\hat{I}(u, v)] = \sqrt{\sum_{y=0}^{Y-1}\sum_{x=0}^{X-1}\Delta I^2(x, y) \sin^2\left(2\pi\mathrm{i}\left(\frac{x}{X}u + \frac{y}{Y}v\right)\right)} \tag{6}$$

Since $\Delta\mathfrak{R}[\hat{I}]$ and $\Delta\mathfrak{I}[\hat{I}]$ both depend on the image uncertainty $\Delta I$, the covariance $\Delta\hat{I}_{\mathfrak{R}\mathfrak{I}}$ has to be taken into account [43, 44]:

$$\Delta\hat{I}_{\mathfrak{R}\mathfrak{I}}(u, v) = \sqrt{\sum_{y=0}^{Y-1}\sum_{x=0}^{X-1}\Delta I^2(x, y) \sin\left(2\pi\mathrm{i}\left(\frac{x}{X}u + \frac{y}{Y}v\right)\right) \cos\left(2\pi\mathrm{i}\left(\frac{x}{X}u + \frac{y}{Y}v\right)\right)} \tag{7}$$

The calculation of $\Delta\mathcal{P}(u, v)$ is straight forward:

$$\Delta\mathcal{P} = 2\sqrt{(\mathfrak{R}[\hat{I}] \cdot \Delta\mathfrak{R}[\hat{I}])^2 + (\mathfrak{I}[\hat{I}] \cdot \Delta\mathfrak{I}[\hat{I}])^2 + 2 \cdot \mathfrak{R}[\hat{I}]\mathfrak{I}[\hat{I}] \cdot \Delta\hat{I}_{\mathfrak{R}\mathfrak{I}}^2} \tag{8}$$

## Noise simulation

To verify the validity of our image transformations and uncertainty calculations, a Monte-Carlo based noise simulation with different test images was carried out.

The uncertainty of each pixel, $\Delta I(x, y) = \sqrt{I(x, y)}$, is assumed as a normal-distributed fluctuation of a repeated measurement $I_k(x, y)$ around the measured intensity $I(x, y) = \frac{1}{N}\sum_{k=0}^{N} I_k(x, y) = \frac{1}{N}\sum_{k=0}^{N}(I(x, y) + \delta I_k(x, y))$ with $\frac{1}{N}\sum_{k=0}^{N}\delta I_k(x, y) = 0$. $N$ is the number of measurements (Fig 1).

The propagation of the image uncertainty $\Delta I(x, y)$ through an arbitrary image operation $I(x, y) \to C(I(x, y))$, $\Delta I(x, y) \to \Delta C(I(x, y))$ is compared to the standard deviation of the Monte-Carlo simulated images $\Delta C_{\mathrm{MC}}$:

$$\Delta C_{\mathrm{MC}} = \sqrt{\frac{1}{N-1}\sum_{k=0}^{N}[C(I(x, y) + \delta I(x, y)) - C(I(x, y))]^2} \tag{9}$$

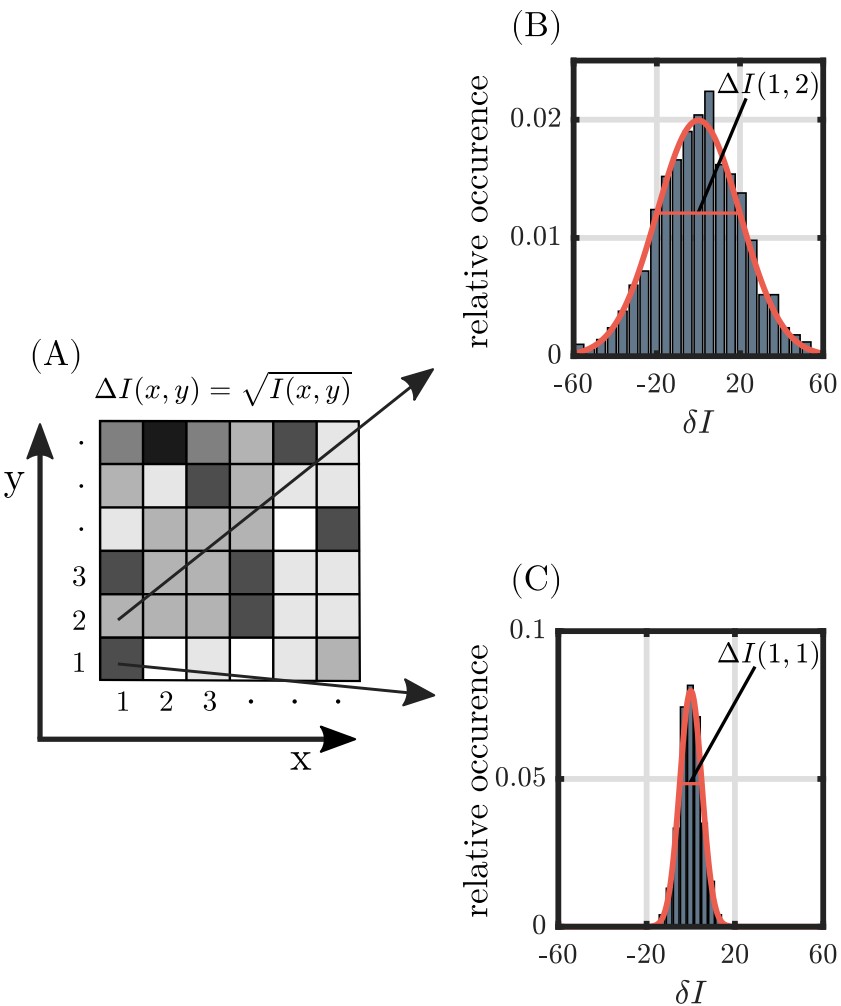

**Fig 1. Schematic outline of the pixel-wise noise simulation.** (A) The photon counting noise results in the standard deviation of each image pixel. In the pixel-wise uncertainty simulation, normal distributions are used to sample the intensity deviation $\delta I(x, y)$ in each pixel, as exemplary shown in (B) and (C) for pixels (1, 2) and (1, 1), respectively. Thereby, high intensity pixels are associated with a large absolute deviation of potential intensity values (B).

The result of Eq 9 is then compared to the calculated uncertainty $\Delta C$ by computing the relative deviation as:

$$\Delta_{\mathrm{MC}} = (\Delta C - \Delta C_{\mathrm{MC}})/\Delta C_{\mathrm{MC}} \tag{10}$$

## Preprocessing and filtering

**Artifact removal.** The computation of the discrete Fourier transform (Eq 1) intrinsically assumes a periodical continuation of the image causing cross-like artifacts to appear in the Fourier domain mainly along the major axis (Fig 2). The magnitude of these artifacts depends on the image intensities near the boundary. Weighting functions in the spatial domain such as the Hamming or Welch window, which gradually reduce the image intensity towards the boundary, are able to remove these artifacts [6, 12, 16, 32, 36, 45]. However, applying window functions is a trade-off between reducing the image content and removing the artifacts in the Fourier domain. To overcome the drawback of weighting functions in the spatial domain, the

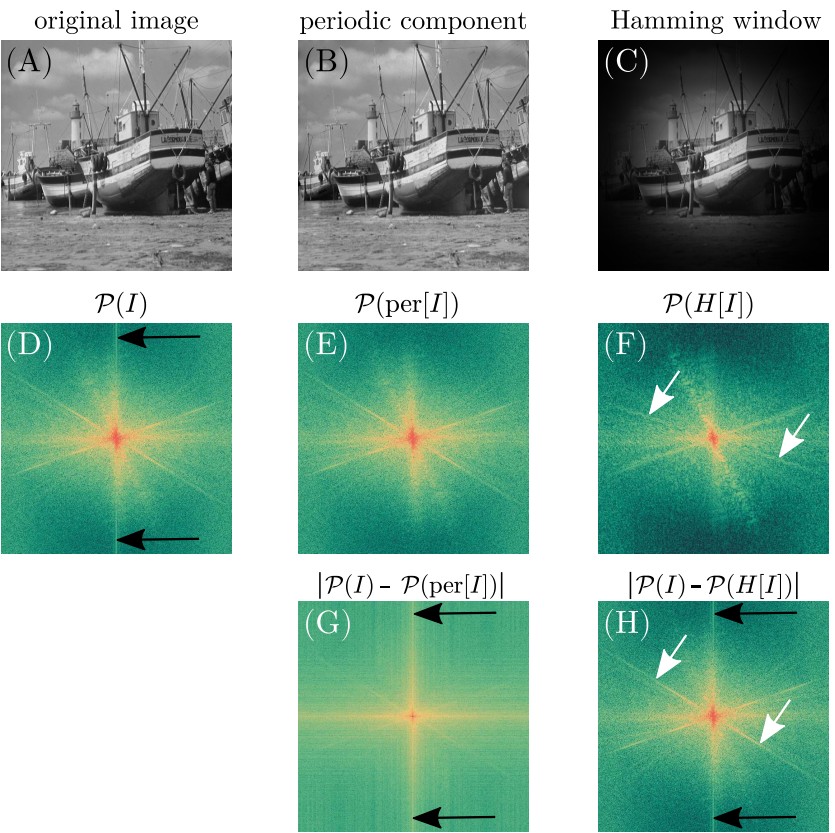

**Fig 2. Effect of spatial domain image filters on the power spectrum.** Images in the spatial domain are shown in greyscale (A-C), whereas the respective power spectra are shown in false colors as well as in logarithmic scale for a better visibility (D-H). Power spectrum values are shifted, such that low frequencies are located in the center. (A) shows the original image *boat* [46]. (B) shows the periodic component of the periodic plus smooth decomposition of Moisan et. al [45]. Apart from small areas near the image boundary, the entire image information of the original image is conserved. However, as shown in (C), the Hamming window reduces the image information gradually towards the image boundary. (D), the discrete Fourier transform of images causes cross-like artifacts (black arrows) to appear in the power spectrum as the image is assumed to be periodical. (E-F), these artifacts disappear in the power spectra if firstly either the periodic decomposition or the Hamming window are applied. Note that the artifacts reappear in (G) and (H) where the absolute difference of the power spectra of the filtered images with respect to the power spectrum of the original image are shown. (G) also reveals that mostly non-directional information is removed from the power spectrum of the original image as only cross-like shapes are pronounced. Other than in (F), where the reduction of image information by the Hamming window also affects the power spectrum, which appears less sharp (white arrows) compared to (D) and (E).

linear decomposition of the image $I(x, y) \rightarrow \text{per}[I(x, y)]$ into a smooth component $s(x, y)$ and a periodic component $p(x, y) = \text{per}[I(x, y)]$ with $I(x, y) = p(x, y) + s(x, y)$ as proposed by Moisan [45] is used here (Fig 2). Since the periodic component per($I$) differs only slightly from $I$ (Fig 2), the effect of the periodic mapping on $\Delta I$ is believed to be negligible. Verification of the effect of the periodic mapping on the image uncertainty is carried out by applying the Monte-Carlo noise simulation and evaluating Eq 9 to three different test images. Classic image processing greyscale test images including the *Lena*, *Cameraman* and *Boat* image were chosen [46]. The *Boat* image serves as example in several figures. The image uncertainty $\Delta I$ is assumed as $\Delta I = \sqrt{I}$, which is indeed an arbitrary choice but since the focus here is on validation only the assumption is reasonable.

The overall mean deviation (Eq 10, $N = 10^5$ iterations) averaged over all images and the entire image range amounts for $\Delta_{\text{MC}} = (0.01 \pm 2.21)\%$. The image uncertainty exceeds the Monte-

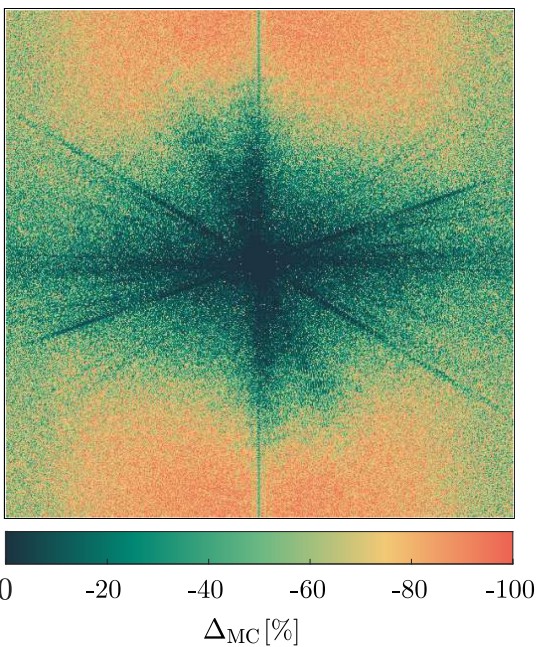

**Fig 3. Relative uncertainty between calculated and simulated uncertainty of the power spectrum.** The relative deviation between the theoretical uncertainty $\Delta\mathcal{P}$ and the Monte-Carlo simulated uncertainty $\Delta\mathcal{P}_{\mathrm{MC}}$, $\Delta_{\mathrm{MC}} = (\Delta\mathcal{P} - \Delta\mathcal{P}_{\mathrm{MC}})/\Delta\mathcal{P}_{\mathrm{MC}}$ of the *Boat* image (Fig 4). Note that the spectrum values are shifted such that low frequencies are located in the origin and that the original image was used here.

Carlo calculated uncertainty at the boundary by $(23 \pm 6)$%. Thus, the original image uncertainty is slightly underestimated due to the uncertainty of the remaining pixels by $-(0.23 \pm 0.61)$%. This deviation is considered as sufficiently low to reasonably assume $\Delta(\mathrm{per}(I)) = \Delta I$. Detailed results for every test image are listed in the supplementary (S1 Table).

**Power spectrum filtering.** The same set of test images is used for validating the calculated uncertainty of the real and imaginary parts (Eqs 5 and 6). The averaged relative deviation (Eq 10) amounts for $(0.00 \pm 0.22)$% for the real and imaginary parts of all images.

Lastly, the Monte-Carlo method was applied on Eq 8 for the same images and same number of iterations. Averaging the relative deviation $\Delta_{\mathrm{MC}}$ over the entire image yields a maximum deviation of $-(59.32 \pm 26.48)$% (S2 Table). However, $\Delta_{\mathrm{MC}}$ correlates well with the relative uncertainty of the power spectrum $\Delta\mathcal{P}/\mathcal{P}$ (Fig 3), which ranges from minor values up to relative uncertainties above 800%. Restricting the area of evaluation to pixels which exhibit a maximum relative uncertainty of 100% increases the accuracy of the uncertainty calculation to $-(8.09 \pm 5.46)$% at maximum (S2 Table, Fig 4). Excluding values with a high relative uncertainty naturally filters the power spectrum, while leaving the back-transformed image relatively unaffected.

In order to find the optimal cut-off value $\delta_{\mathrm{cut}}$, images with known ground truth have to be used. Since fibrous images are in the scope of this work, Monte-Carlo generated greyscale fiber images serve as test images for determining the optimal cut-off value.

## Monte-Carlo image generation

Besides the Monte-Carlo simulation for noise simulation, another Monte-Carlo routine was implemented for the generation of images containing fibers with a known angular distribution.

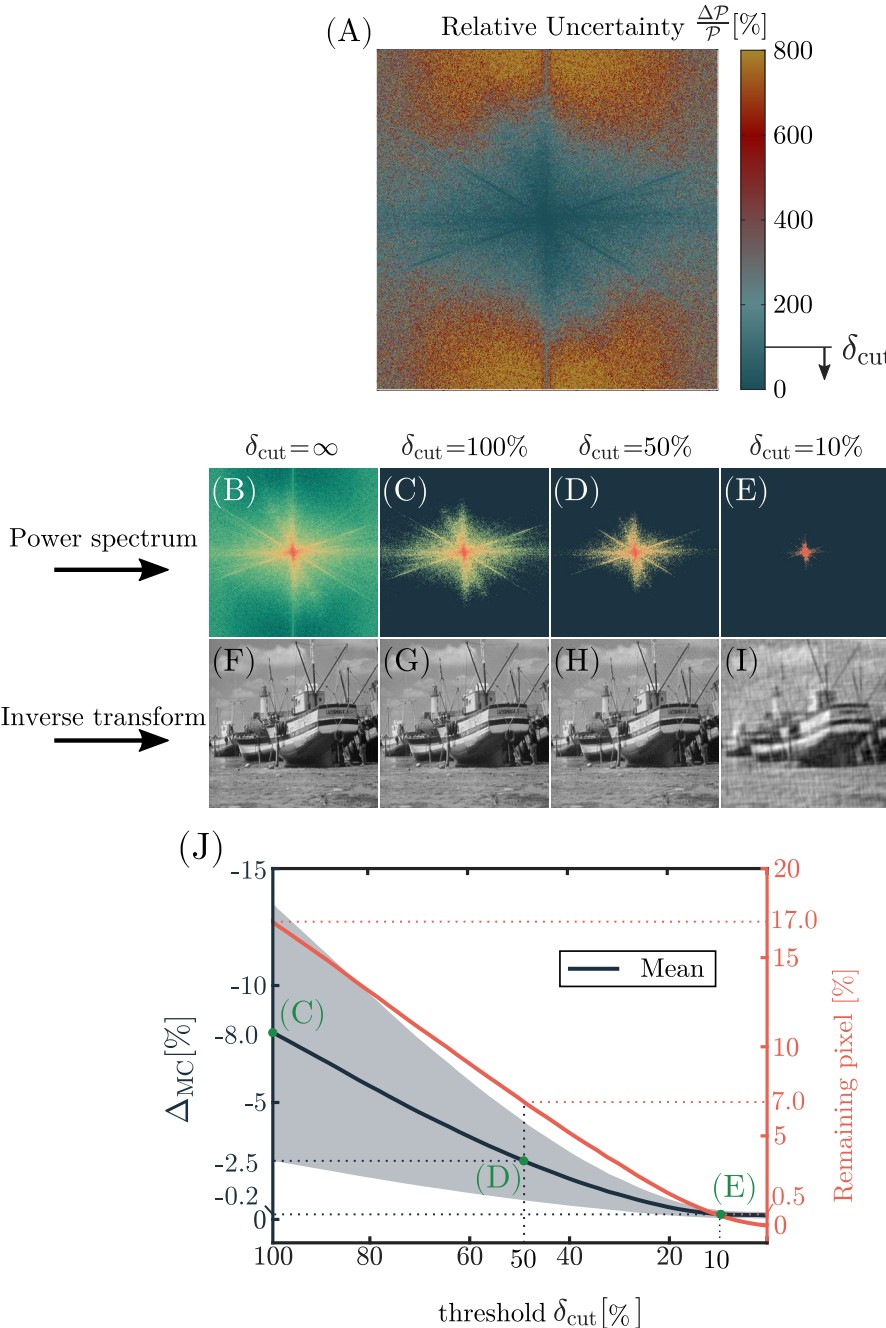

**Fig 4. Relative uncertainty and filtering of the power spectrum.** The relative uncertainty (A) of the power spectrum increases with distance from the origin to values above 800%. A filter mask for the power spectrum is achieved from excluding relative uncertainties above $\delta_{\mathrm{cut}}$. (B-E) show the filtered power spectra for different $\delta_{\mathrm{cut}}$ with the respective inverse Fourier transformations in (F-I). The number of remaining pixels after filtering as well as the error of the uncertainty calculation compared to the Monte-Carlo simulation $\Delta_{\mathrm{MC}}$ is shown in (J). The mean error, its standard deviation (*grey* shaded areas) as well as the number of remaining pixels gradually decrease towards a lower threshold $\delta_{\mathrm{cut}}$. Note that even if 99.5% of the pixels are excluded, the contours of the major image components are still visible (E).

A single fiber is defined by its width, length, orientation and location. While several publications dealt with the generation of binary fiber images [12, 34] we focus on the generation of random greyscale intensity images including noise knowing that the contrast of the fibers influence their contribution within the power spectrum [36]. Since our goal here is to find the

optimal evaluation parameters for a large variety of realistic images similar to the SHG images of dermal collagen that are used later on, the usage of binary images is not reasonable. In addition, our approach aims on evaluating images without any preprocessing, thus the test images should cover different image qualities.

The orientation of the fibers is sampled from a semi-circular $\pi$-periodic von-Mises distribution:

$$P(\theta; \bar{\theta}, k) = \frac{1}{\pi I_0(k)} e^{k \cdot \cos(2(\theta - \bar{\theta}))} \tag{11}$$

with the dispersion parameters $k$ and $\bar{\theta}$ defining the width and the center of the distribution. $I_0(k)$ is the modified zero order bessel function $I_0(k) = \frac{0}{\pi} \int_0^\pi \cos(x) \mathrm{d}x$. A rejection sampling algorithm is used to sample fiber angles. The intensity of each sampled fiber with certain width and aspect ratio is added to the existing image to obtain a greyscale image. The image is then smoothed using a gaussian kernel with standard deviation of two to slightly dissolve sharp edges. After that, intensities are scaled such that the maximum intensity is equal to the maximum intensity of a 16-bit intensity image. Width, aspect ratio and dispersion $k$ of the fibers affect the accuracy of the image processing algorithm [34]. The minimum fiber width amounts for one pixel as the maximum fiber width is confined by the image size and the maximum allowed aspect ratio. As images at $512 \times 512$ pixels are generated, a maximum aspect ratio of 45 and a maximum fiber width of 10 are chosen to still allow for an effective placement of the fibers within the image boundaries. A minimum aspect ratio of 10 was proposed by Marquez et. al [6] to allow for a reasonable evaluation of orientation. As we are generating greyscale fiber images with overlapping fibers we choose a minimum aspect ratio of 15. Dispersed as well as aligned fiber networks are achieved by sampling $k$ within [0.01, 5].

To enforce different image qualities, we introduce a noise factor. The noise factor ranges from a minimum of 0, which corresponds to a completely denoised image up to a maximum of 1 which corresponds to random speckle noise which can value up to half of the maximum intensity of the image.

## Angular distribution generation

Quantifying the angular orientation of the fibers by means of mean angle and fiber dispersion requires the calculation of the total intensity of each angle of the filtered power spectrum. This is realized by using a radial summation. The total intensity of a certain angle is given by the sum over all pixels of the power spectrum, that are included within the angular range $[-\delta\theta, +\delta\theta]$. The contribution of each pixel is weighted by the percentage of area, which is included in $[-\delta\theta, +\delta\theta]$.

$$\mathcal{I}(\theta) = \sum_{\theta \in [-\delta\theta, \delta\theta]} \sum_r \mathcal{P}'(\theta, r) \cdot w_\theta(\theta, r) \qquad \text{with} \qquad \frac{1}{X \cdot Y} \sum_\theta \sum_r w_\theta(\theta, r) = 1 \tag{12}$$

The uncertainty propagates as:

$$\Delta\mathcal{I}(\theta) = \sqrt{\sum_{\theta \in [-\delta\theta, \delta\theta]} \sum_r (\mathcal{P}'(\theta, r) \cdot w_\theta(\theta, r))^2} \tag{13}$$

The calculated intensity is normalized such that $\sum_\theta \mathcal{I}(\theta) = 1$. An issue that is faced here are high intensity pixels close to the center of the power spectrum, which do not provide any directional information. The intensity of these pixels is several magnitudes higher than the intensities of interest, which causes artifacts in the angular distribution. A sensitivity analysis

of a set of test images showed that zeroing pixel with a distance smaller than 3 pixels from the center is sufficient to remove these artifacts (S1 Fig).

Additionally, a modified intensity is defined, which exploits the anisotropy of the introduced filter. Let $N_{\Delta\theta}$ be the number of non-zero pixels within the angular range $\Delta\theta$ of the filtered power spectrum $\mathcal{P}'$. The modified intensity then reads as:

$$\mathcal{I}(\theta) = N_{\Delta\theta}^{\alpha} \cdot \sum_{\theta \in [-\delta\theta, \delta\theta]} \sum_{r} \mathcal{P}'(\theta, r) \cdot w_{\theta}(\theta, r), \tag{14}$$

with the uncertainty given by:

$$\Delta\mathcal{I}(\theta) = N_{\Delta\theta}^{\alpha} \cdot \sqrt{\sum_{\theta \in [-\delta\theta, \delta\theta]} \sum_{r} (\mathcal{P}'(\theta, r) \cdot w_{\theta}(\theta, r))^2}, \tag{15}$$

where $\alpha$ defines the impact of $N_{\Delta\theta}$. A value of $\alpha = -1$ corresponds to an averaged intensity, whereas $\alpha > 0$ amplifies the number of remaining pixels within the given angular range. The ideal choice of $\alpha$ strongly depends on the chosen cut-off value $\delta_{\text{cut}}$ for the power spectrum. For example in case of a low cut-off value, the filtered power spectrum $\mathcal{P}'$ might exhibit a high anisotropy, where $\alpha > 1$ might lead to a more accurate result compared to the unweighted sum. Considering a very high cut-off value, $\mathcal{P}' = \mathcal{P}$ holds and the number of summed pixels should not have any influence. Thus $\alpha = -1$ might be the value of choice. A somewhat similar approach was used by Polzer et. al [36], which used a factor to empower the entire intensity distribution $\mathcal{I}(\theta)$.

To find the optimal parameter set $[\theta_{\text{cut}}, \alpha]$, a two parameters optimization algorithm is applied on $N = 10^4$ Monte-Carlo generated images. For this purpose we use the MATLAB-implemented Nelder-Mead simplex method (*fminsearch*) [47]. To account for different types of images, fiber properties, namely width, aspect ratio and dispersion as well as the noise factor are randomly sampled from the respective interval specified before. We use the mean squared difference (MSD) between the calculated orientation distribution $\mathcal{I}_{\text{MC}}(\theta)$ and the prescribed orientation distribution $\mathcal{I}(\theta)$ as objective function, which is minimized:

$$\text{MSD} = \frac{1}{N-1} \sum_{i=0}^{N} \text{SD}_i \qquad \text{with} \qquad \text{SD}_i = \sum_{\theta} (\mathcal{I}_i(\theta) - \mathcal{I}_{i,\text{MC}}(\theta))^2 \tag{16}$$

Termination was enforced as soon as the difference between consecutive iterations of both parameters was smaller than $10^{-3}$. MATLAB uses the same termination criterion for both parameters. Therefore, we scaled $\alpha$ by a factor of 0.1 to match the magnitude of $\delta_{\text{cut}}$. A subset of $10^3$ images was evaluated prior optimization on a $10 \times 10$ grid with $\alpha \in [-1, 8]$ and with $\delta_{\text{cut}} \in [1, 100]\%$ to ensure that the calculated minimum is global and to get an estimate for the starting values of each parameter. We estimated that $\alpha = 2$ and $\delta_{\text{cut}} = 2\%$ provide a suitable set of starting values.

### Angular distribution quantification

**Conventional approach.** The common approach for quantifying the mean orientation and dispersion is to fit a semi-circular von-Mises function (Eq 11) to the distribution $\mathcal{I}(\theta)$. This approach provides accurate results in case of aligned fibers but looses accuracy in case of isotropic distributions [34]. Additionally, the use of an arbitrary averaging range to smooth the angular orientation distribution prevents a meaningful and objective interpretation of the data in terms of e.g. the number of dominant fiber directions.

**New fitting approach.** To find an objective evaluation procedure which reliably can deal with unprocessed data, fitting the cumulative distribution function (CDF) $\mathcal{C}(\theta)$ is quite appealing, since the data are naturally smoothed:

$$\mathcal{C}(\theta) = \sum_{\theta'=-90°}^{\theta} \mathcal{I}(\theta') \tag{17}$$

The uncertainty of $\Delta\mathcal{C}$ is given by a quadratic summation:

$$\Delta\mathcal{C}(\theta) = \sqrt{\sum_{\theta'=-90°}^{\theta} (\Delta\mathcal{I}(\theta'))^2} \tag{18}$$

Since the choice of the starting angle of the summation (Eq 17) is arbitrary, the uncertainty $\Delta\mathcal{C}$ must be independent from $\theta$. Thus, the maximum uncertainty of $\mathcal{C}(\theta)$ is set as uncertainty for all angles. A peak in the angular distribution corresponds to a step in the cumulative distribution, which approaches a first order polynomial for isotropic distributions. To model the cumulative orientation distribution a sigmoid function is chosen:

$$S(\theta; b, \bar{\theta}) = \frac{1}{1 + e^{-b \cdot (\theta - \bar{\theta})}} \tag{19}$$

The advantage of fitting a von-Mises distribution is its semi-circularity, namely $P_{\mathrm{VM}}(\theta) = P_{\mathrm{VM}}(\theta + 180°)$. Since the cumulative distribution function is monotonically increasing, the corresponding condition reads as $\mathcal{C}(\theta) = \mathcal{C}(\theta - 180°) + 1 = \mathcal{C}(\theta + 180°) - 1$. In order to meet this condition, Eq 19 is modified by adding neighboured sigmoid functions accounting for the added offsets:

$$S_{\mathrm{circ}}(\theta; b, \bar{\theta}) = S(\theta; b, \bar{\theta}) + S(\theta + 180°; b, \bar{\theta}) - S(180°; b, \bar{\theta}) + S(\theta - 180°; b, \bar{\theta}) - S(-180°; b, \bar{\theta}) \tag{20}$$

## Validation

**Comparison to band-pass filtering.** In the following we refer to our method as AF method (adaptive filtering method) for convenience. To capture the performance of the implemented procedure we compare the AF method to the state of the band-pass method, which has been proven to provide more accurate results than other methods [34]. In order to observe the influence of fiber and image properties on the accuracy of both methods, we created a dataset, where each property is altered separately using the introduced Monte-Carlo method for generating greyscale images. The following sets of values were used for creating the image dataset:

$$\text{width} = \{1, 5, 10\}, \qquad \text{AR} = \{15, 30, 45\}, \qquad \text{noise factor} = \{0, 0.5, 1\} \tag{21}$$

$$\text{and} \quad k = \{0.01, 0.25, ..., 5\} \tag{22}$$

In order to enable a reasonable statistical evaluation, 20 images with random mean fiber orientation $\bar{\theta}$ are created for each category, which adds up to a total of $N = 3^3 \cdot 21 \cdot 20 = 11,340$ images. Each image is evaluated using the AF method using the optimal evaluation parameters $\delta_{\mathrm{cut}} = 2.1\%$ and $\alpha = 1.5$ and using the band-pass method of Morrill et. al by applying their provided software *FiberFit*. We follow their recommendations of the upper and lower cutoff

parameter $t = 2$ and $t = 32$ yielding a lower cutoff frequency of 8 and an upper cutoff frequency of 128. The resulting evaluation parameters, mean orientation $\bar{\theta}$ and dispersion coefficients $b$ and $k$ are compared to the reference parameters. To account for Monte-Carlo sampling errors, reference mean orientation and distribution parameter $k$ are obtained from fitting the frequency distribution of sampled fiber angles. The reference dispersion parameter $b$ is obtained from fitting the cumulative frequency distribution of sampled fiber angles using the modified sigmoid function (Eq 20). Mean orientations $\bar{\theta}_{FF}$ of the FiberFit software were inverted and shifted to match our coordinate system $\bar{\theta}'_{FF} = 180° - \bar{\theta}_{FF}$. Additionally, angles exceeding the interval [0°, 180°] were shifted by 180°. Comparisons to the reference parameters are performed by considering the absolute error of the mean orientation angle angle $\Delta\bar{\theta}$ and the absolute relative error of the dispersion parameters $\Delta b_{\mathrm{rel}}$ and $\Delta k_{\mathrm{rel}}$. A small fraction of images were found to exhibit large relative errors $\Delta b_{\mathrm{rel}}$ and $\Delta k_{\mathrm{rel}}$. Those are classified as outliers if they are exceeding three times the interquartile range of the respective distribution.

**Statistics.** A paired t-test is used to calculate the level of significance (p-value) between the error of both evaluation methods for the same subset of images in terms of width, aspect ratio, dispersion and noise. If a value is classified as outlier, the corresponding value of the pair is excluded for p-value calculation. An unpaired t-test is used for p-value calculations among groups with different noise factor. Significance levels of 0.05, 0.01 and 0.001 are marked as (\*, \*\*, \*\*\*), respectively.

**Application on experimental data.** Finally, we checked the applicability of the AF method on in vivo experimental data. We use SHG-images of dermal collagen that were recorded using a CE-certified multi-photon microscope (DermaInspect) for in vivo applications, which was developed in collaboration with Jenlab GmbH (Jena, Germany). The SHG signal was captured using an excitation wavelength of 820 nm together with a specific band-pass filter (410 ± 10 nm, AQ 410/20m-2P, Chroma Technology Corp., Bellows Falls, VT). A scan time of 13 s and image dimensions of 512 × 512 pixels with a 220 × 220 $\mu m$ field of view were chosen as image acquisition parameters. For a detailed description of the microscope refer to [48–50]. In total, ten SHG images of dermal collagen were taken from ten different volunteers at a depth of 30 $\mu m$ under the basal membrane located at the cheek.

This study was conducted according to the recommendations of the current version of the Declaration of Helsinki and the Guideline of the International Conference on Harmonization Good Clinical Practice, (ICH GCP). In addition, this study was approved and cleared by the institutional ethics review board (Beiersdorf AG, Hamburg, Germany). Written informed consent was obtained from each volunteer.

To get the reference angular distribution we manually trace the collagen fibers. Statistical variance is achieved by tracing each image three times in random order. The angular orientation distribution is achieved from adding up the length of each fiber being oriented along a certain angle in 1° steps. A smooth distribution is obtained by filtering the data using a Gaussian kernel with a standard deviation of 1°. Reference parameters $\bar{\theta}_m$ and $b_m$ are obtained by fitting the modified sigmoid function (Eq 20).

## Results

### Angular distribution generation

Fig 5 shows the result of the optimization procedure of the evaluation parameters $\delta_{\mathrm{cut}}$ and $\alpha$. Apart from a minor fraction of outliers, data points of minimal difference spread within $\delta_{\mathrm{cut}} < 4\%$ and $0 < \alpha < 3$. Since the frequency distributions $p(\delta_{\mathrm{cut}})$ and $p(\alpha)$ are not normally distributed, the median value of both parameters is considered as estimation of the overall minimum

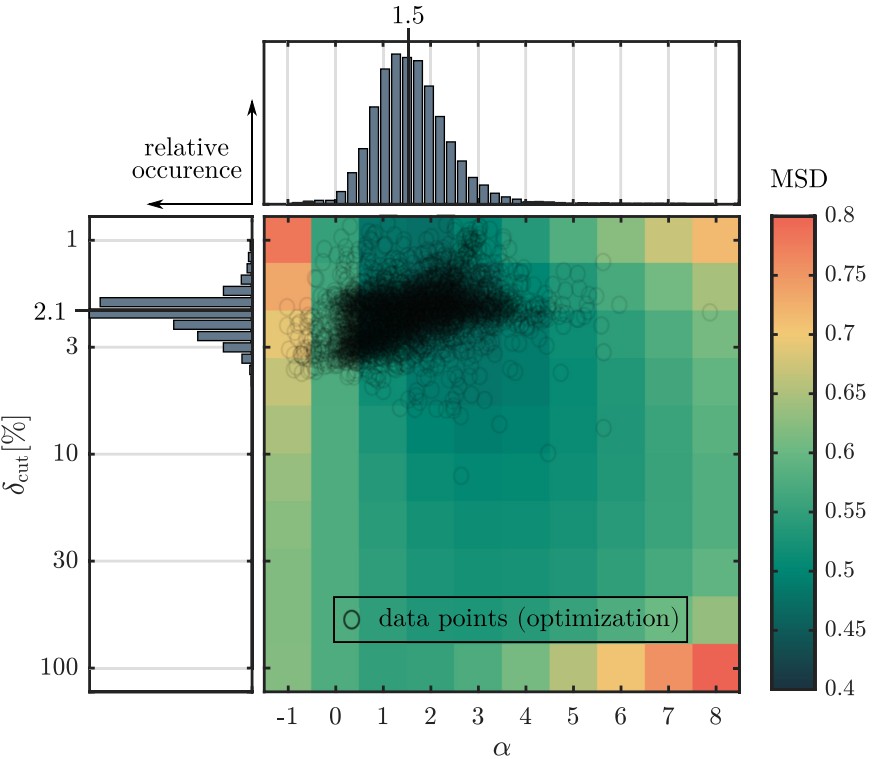

**Fig 5. Effect of evaluation parameters $\delta_{cut}$ and $\alpha$ on the error.** False colours indicate the mean squared difference MSD between the computed angular distributions and their reference on a subset of $N = 10^3$ images for a discrete set of $10 \times 10$ parameter values. Data points represent the optimal parameter set for each image ($N = 10^4$) based on the exploited two parameters Nelder-Mead optimization algorithm which used the squared difference between reference and calculated orientation distribution as objective function. Histograms show the frequency distribution of parameter $\delta_{cut}$ and $\alpha$. The median values of both distributions($\delta_{cut} = 2.1\%$, $\alpha = 1.5$) were identified as optimal evaluation parameters. The mean number of iterations, until the termination criterion was met, is $23 \pm 9$. Only 0.2% of the images reached the maximum number of iterations, which was set to 100.

($\delta_{cut} = 2.1\%$, $\alpha = 1.5$). The mean squared difference minimizes for $\alpha$ in [1, 3] and for $\delta_{cut} <$ 10% indicating that a global minimum is considered. Subsequent calculations using the AF method are performed with $\delta_{cut} = 2.1\%$ and $\alpha = 1.5$. Median values of the mean squared difference as a function of fiber dispersion $k$, image quality (NF) and fiber geometry (width, aspect ratio) are provided in the supplement (S2 Fig).

## Validation

**Comparison to band-pass filtering.** The overall error of the mean fiber orientation, $\Delta\bar{\theta}$ amounts for $(2.2 \pm 1.8)°$ and $(1.8 \pm 1.4)°$ ($p < 0.001$) for the band-pass and the AF method, respectively (Table 1). Note that for calculating the overall error $\Delta\bar{\theta}$ highly dispersed fiber

**Table 1. Overall error of mean orientation and dispersion determined by the band-pass method and by the AF method.** Note that $f$ indicates the percentage of data points that were classified as outlier.

| | $\Delta\bar{\theta}[°](k > 1)$ | $\Delta b_{rel}\vert\Delta k_{rel}[\%]$ | $f_{\Delta b\vert\Delta k}[\%]$ | $R^2$ |
|---|---|---|---|---|
| band-pass method | $2.3 \pm 2.1$ | $33.9 \pm 26.5$ | 0.73 | $0.78 \pm 0.24$ |
| AF method | $1.9 \pm 1.5$ | $13.2 \pm 12.7$ | 0.76 | $0.99 \pm 0.00$ |
| $p$-value | $< 0.001$ | $< 0.001$ | | $< 0.001$ |

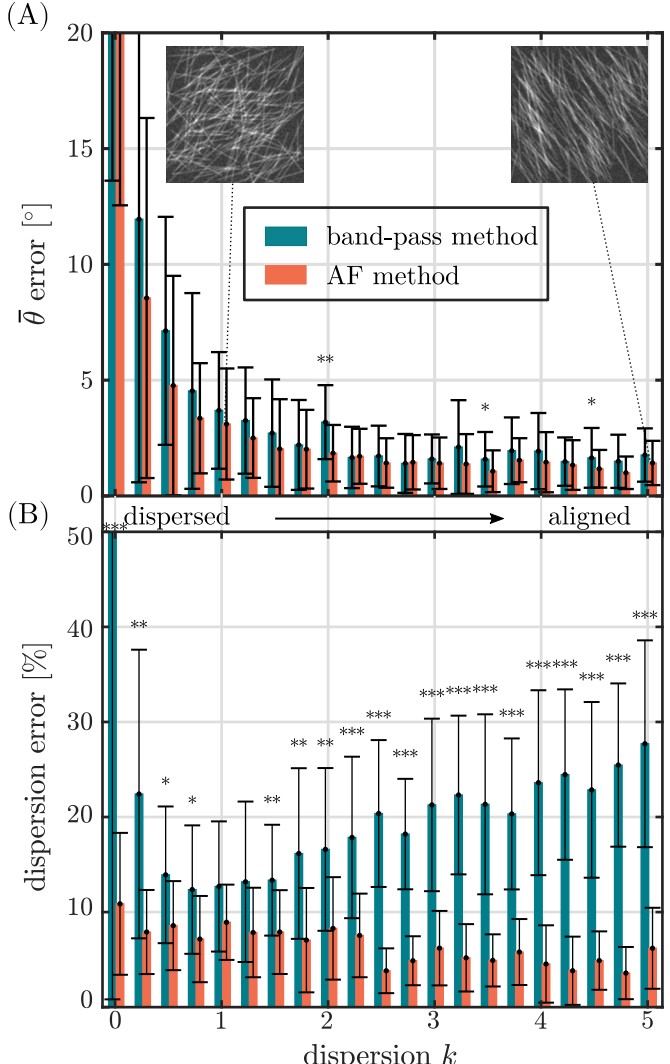

**Fig 6. Error of the band-pass method and the AF method vs. the dispersion $k$ of the fiber network.** Fibers with a width of 5, aspect ratio of 30 and an image noise factor of 1 were considered here. (A) shows the error of the mean fiber orientation $\Delta\bar{\theta}$. Inlets show exemplary Monte-Carlo generated greyscale fiber images from respective distributions. (B) shows the relative error of the fiber dispersion parameters $\Delta k_{rel}$, $\Delta b_{rel}$. Sample greyscale images, that were simulated by the implemented Monte-Carlo method are shown for $k = 0.01$ and $k = 5$. ($^{*}p < 0.05$, $^{**}p < 0.01$, $^{***}p < 0.001$).

networks ($k < 1$) were excluded from statistical analysis since they do not feature a mean orientation. The overall mean error of the dispersion parameters $\Delta k_{rel}$, $\Delta b_{rel}$ amounts for $(33.9 \pm 26.5)\%$ and $(13.2 \pm 12.7)\%$ ($p < 0.001$) using the band-pass and the AF method, respectively. It was found that for both methods the error of the mean fiber orientation $\Delta\bar{\theta}$ mainly depends on the degree of the alignment of the fiber network (Fig 6A) and is rather independent from the fiber geometry or image quality (not shown). The error of both methods reduces towards an increasing degree of fiber alignment.

The error of the dispersion parameters $\Delta b_{rel}$ and $\Delta k_{rel}$ is found to strongly depend on the degree of alignment, fiber geometry and image noise (Figs 6B and 7). With increasing the degree of alignment, the fiber dispersion error of the AF method, $\Delta b_{rel}$, continuously decreases from a maximum error of 10.9% at $k = 0.01$ to a plateau of $\sim5\%$ for $k > 2.2$. The

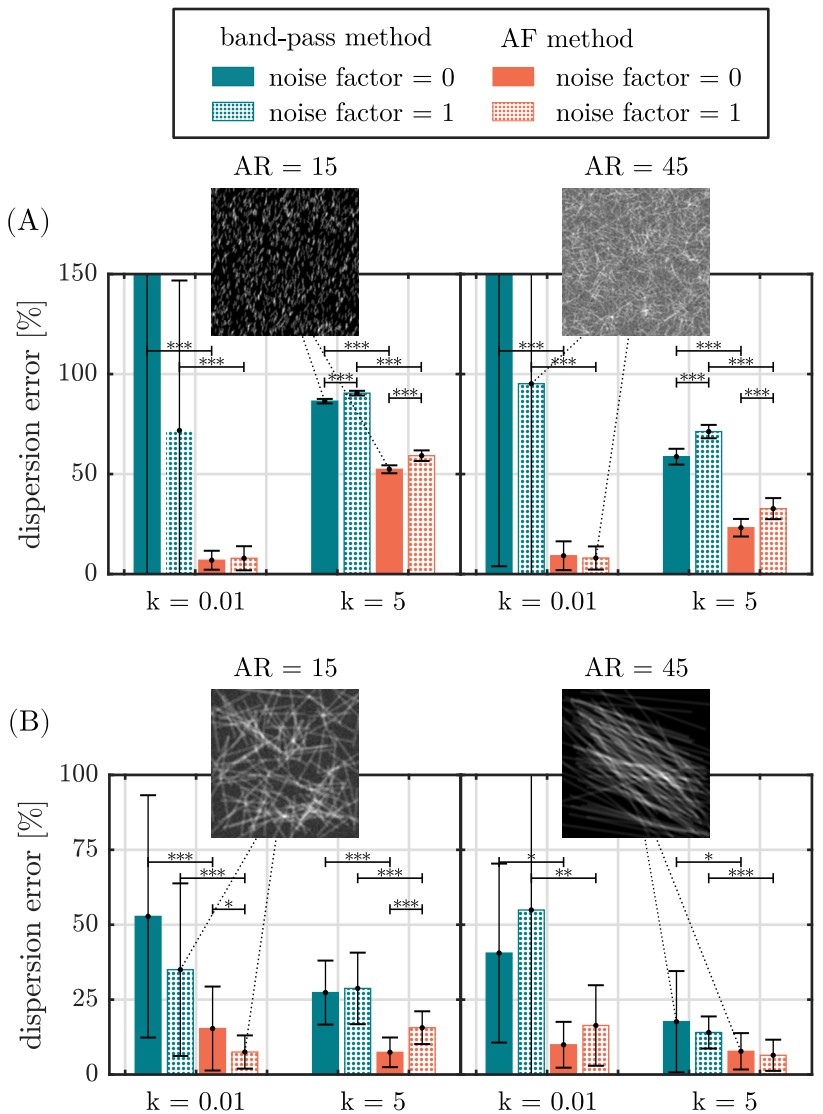

**Fig 7. Error of the dispersion parameter of the band-pass method and the AF method for different fiber geometries, dispersions and noise factors.** (A) shows the error of the dispersion parameter for fibers with a width of 1 pixel for aspect ratios 15 and 45 as well as noise factors 0 and 1. (B) shows the same as in (A) but for fibers with a width of 10 pixels. Each error is given for a dispersed ($k = 0.01$) and an aligned fiber network ($k = 5$). Inlets show exemplary Monte-Carlo generated greyscale fiber images from respective distributions. ($^*p < 0.05$, $^{**}p < 0.01$, $^{***}p < 0.001$).

fiber dispersion error of the band-pass method, $\Delta k_\mathrm{rel}$, exhibits a large error at $k = 0.01$ $\Delta k_\mathrm{rel} = 101.4\%$, which is first reduced to a minimum error of $\Delta k_\mathrm{rel} \sim 12\%$ around $k \sim 1$ but then is increased again towards aligned fiber networks to an error of 27.7% at $k = 5$ (Fig 6B). Except for $k = 1$ and $k = 1.25$ the error of both methods shows a significant ($p < 0.05$), for most values of $k$ even highly significant ($p < 0.001$) difference.

Other than the error of the mean fiber orientation, the error of the dispersion parameter strongly depends on the fiber geometry, image noise and choice of evaluation method (Fig 7). An at least significant decrease ($p < 0.05$) in error was found for every group if the image was evaluated with the AF method.

A decreased fiber width strongly increases $\Delta k_\mathrm{rel}$ for every combination of aspect ratio, dispersion and noise. In case $\Delta b_\mathrm{rel}$, the increase in error can be noted for aligned networks only.

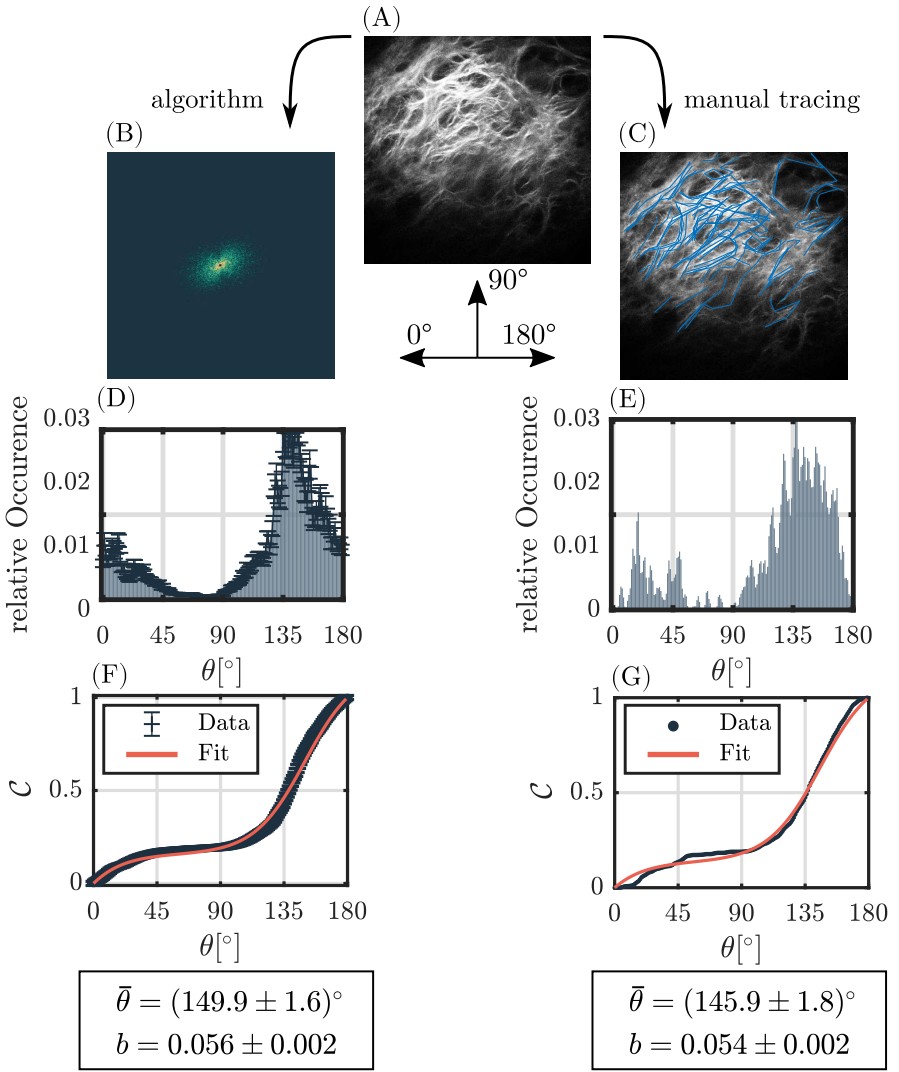

**Fig 8. Evaluation example of an in vivo SHG-image of dermal collagen.** Left column ((B),(D),(F)): Evaluation steps of the implemented image processing algorithm. Right column ((C),(E),(G)): Manual fiber tracing of fibers, which serves as ground truth. (A) shows the original SHG-image as measured with the multi-photon microscope. (B), the filtered power spectrum for $\delta_{cut}$ = 2.1%. (C), manually traced fibers. (D) and (E) show the resulting angular orientation distributions. For (D), $\alpha$ = 1.5 was used. (F) and (G) show the respective cumulative orientation distributions and the fit parameters obtained from sigmoid fitting.

A raised noise factor significantly ($p < 0.001$) increases $\Delta k_{rel}$ and $\Delta b_{rel}$ for aligned networks of fibers with a width of 1. Additionally, a significant ($p < 0.05$) decrease and a highly significant increase ($p < 0.001$) in $\Delta b_{rel}$ was found for thick, short fibers (width = 10, AR = 15).

**Application on experimental data.** Our implemented algorithm was applied on an exemplary set of ten in vivo recorded SHG images of dermal collagen. Parameters of the cumulative angular distribution $\bar{\theta}$ and $b$ were calculated using the AF method and compared to reference parameters $\bar{\theta}$ and $b$ achieved from manual fiber tracing (Fig 8, S1 Dataset). The absolute mean error between calculation and manual segmentation for the mean orientation amounts for $\Delta\bar{\theta} = (6.0 \pm 4.0)^{\circ}$, whereas the mean relative error of the fiber dispersion is $\Delta b_{rel} = (9.3 \pm 12.1)$%. The mean coefficient of determination was $R^2 = 0.99 \pm 0.01$.

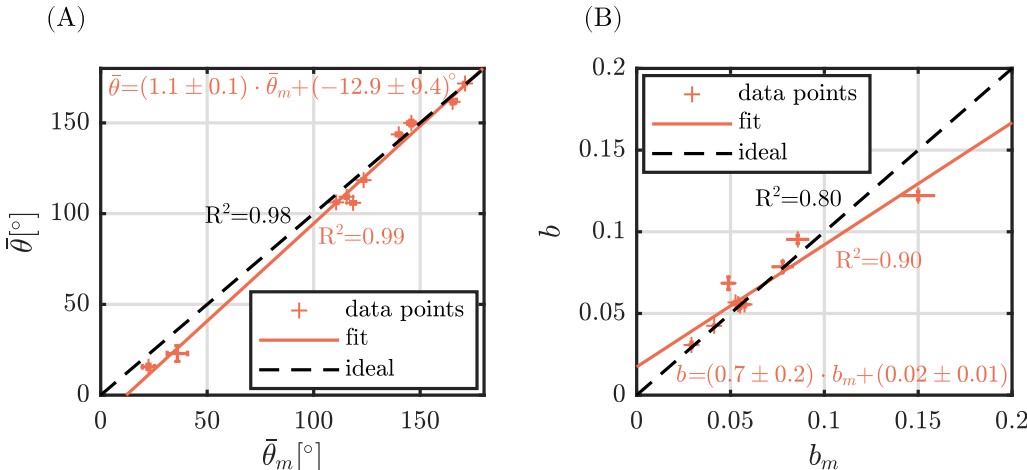

**Fig 9. Calculated angular distribution parameters vs. reference distribution parameters for the experimental data.**
(A) Mean fiber orientation $\bar{\theta}$ vs. mean fiber orientation $\bar{\theta}_m$ achieved from manual segmentation. (B) Fiber dispersion $b$ vs. fiber dispersion $b_m$ achieved from manual segmentation. $R^2$ values are given for the fitted curve (solid, orange) as well as for the ideal curve (dashed, black) with a slope of one. Errorbars represent 95% confidence bounds of fitted parameters.

Value pairs of calculated and reference parameter were fitted using a first order polynomial (Fig 9). High Pearson correlations were found for both parameters, namely $R^2 = 0.99$ and $R^2 = 0.90$ for the mean orientation angle and dispersion, respectively. Calculated Pearson correlations with respect to an ideal calculation (slope equal to one) amount for $R^2 = 0.98$ and $R^2 = 0.8$.

## Discussion

We report on a robust method to quantify the angular distribution of fibers in noisy greyscale fiber image. The whole image processing procedure covers: the application of the periodic decomposition to remove cross-like artifacts in the Fourier domain, Fourier filtering by only permitting values below a certain relative uncertainty $\delta_{\text{cut}}$, computation of the angular distribution by weighting the number of pixels with $N^\alpha$ and quantification of the mean angle and dispersion by fitting a modified sigmoid function to the cumulative orientation distribution.

In comparison to conventionally used window functions like in [12, 16, 32, 36], the periodic decomposition has the advantage of conserving almost the entire image information while completely removing artifacts in the Fourier domain, as shown in Fig 2. Therefore, we omit a quantitative analysis. The Monte-Carlo noise simulation revealed that the effect on the uncertainty can be neglected since the periodic mapping only has significant effects on the image boundary only.

Filtering the power spectrum by excluding values above a predefined relative uncertainty allows for the definition of an adaptive filter. Optimal evaluation parameters were identified by applying a two parameter Nelder-Mead optimization, which succeeded for 99.8% of the images. A maximum cut-off error of $\delta_{\text{cut}} = 2.1\%$ and a weighting factor of $\alpha = 1.5$ was calculated, while the non-locality of the optimum was ensured. Polzer et. al introduced a similar weighting factor which raises the entire intensity distribution $I(\theta)$ [36]. However, the optimum value of their weighting factor seems to suffer from large fluctuations, whereas the optimal value of $\alpha$ is stable throughout the degree of alignment of fiber networks and throughout different fiber properties and image noise (S2 Fig).

Previously used filtering methods like the bandpass method [6, 34, 36] work with rotationally symmetric filters, which disregard the anisotropy of the signal. The derivation of optimal band-pass range is based on the fiber diameter [6]. With the AF method we apply a relative uncertainty criterion without any assumptions on the underlying fiber properties. The optimal value of $\delta_{\text{cut}}$ is found to be completely independent from the chosen fiber geometries, fiber disperion and image noise (S2A, S2C, S2E and S2G Fig). As a consequence, the fiber dispersion can be quantified with a significantly lower error for completely dispersed and highly aligned fiber networks in comparison to the band-pass method (Fig 6B). Especially the quantification of highly dispersed fiber networks ($k < 1$) is much more reliable using the AF method, whereas the dispersion calculated by the band-pass method suffers from large errors ($\Delta k_{\text{rel}} > 100\%$) (Figs 6B and 7). This is in accordance with the observations of Morrill et. al [34], who measured an error of $\sim 30\%$ for $k \sim 0.2$ using binary Monte-Carlo images.

If we use the AF method in conjunction with a von-Mises fit of the angular orientation distribution, large errors of the dispersion coefficient can be noted towards dispersed fiber networks (S3 Fig). Contrary to the band-pass method, $\Delta k_{\text{rel}}$ rapidly decreases towards aligned networks to a level of $\sim 10\%$. This effect is solely related to the adaptive filter and the weighted radial summation of the AF method, whereas the sigmoid fit ensures a reliable dispersion parameter estimation for dispersed networks.

Fibers with a width of 1 somehow represent an exception in comparison to the images containing fibers with width $> 1$ pixel. Regarding the result of the optimization of $\alpha$ with respect to different image and fiber properties (S2 Fig), it can be seen that $\alpha$ is quite stable throughout the dispersion $k$, noise factor and different fiber geometries. $\alpha$ fluctuates within the interval $[1, 2]$ except for fibers with a width of 1 pixel, where the median value $\alpha$ amounts for $\alpha_{\text{width}=1} = 2.4$. The application of the gauss-filter to dissolve sharp edges might even reduce the true fiber dimension down to a size which is near the resolution limit of power spectrum based methods resulting in large deviations of the dispersion for highly aligned fiber networks.

Using the Monte-Carlo approach to create artificial fibrous images for validation purposes is a common tool. However, it is rather difficult to draw a comparison to the accuracy of other methods found in the literature since mostly a low quantity of binary images was investigated [12, 34]. For example, Morrill et. al measured an overall error of $(0.71 \pm 0.43)°$ for the mean orientation and $(7.4 \pm 3.0)\%$ for the fiber dispersion using binary fiber images, whereas we measured errors of $(2.3 \pm 2.1)°$ and $(33.9 \pm 26.5)\%$ [34]. The use of greyscale images will generally result in a lower accuracy compared to the evaluation of binary images, since the superimposition of fibers might generate intensity deviations in the power spectrum.

The AF method was applied to in vivo SHG images of dermal collagen. The multi-photon images that were used provide a sufficiently high image intensity $I(x, y)$ (16-bit), which comes along with a sufficiently low relative intensity error $\Delta I/I = 1/\sqrt{I}$. Therefore, $\delta_{\text{cut}} = 2.1\%$ provides a reasonable filtering value, which results in an accurate quantification of the angular orientation distribution in terms of mean fiber orientation and fiber dispersion.

## Conclusion

The proposed adaptive filter method modifies common Fourier methods at different stages, namely artifact removal in the Fourier domain, filtering of the power spectrum and quantification of the angular signal.

The adaptive filter conserves the anisotropy of the angular signal in the Fourier domain, which ensures a stable error for disordered as well as highly aligned fiber networks. Using the cumulative distribution naturally averages the data, which spares out any averaging of the angular distribution. The high mean goodness of the fit $R^2 = 0.99$ which was measured for

both, Monte-Carlo images and SHG-images, indicates that the modified sigmoid function provides a perfect model of the cumulative distribution function.

The adaptive filtering method was found to be a reliable and accurate tool for quantifying the angular orientation distribution in fibrous SHG images of dermal collagen, even for images suffering from a low image quality. Aside from its benefits concerning accuracy and reliability, the AF method considers measurement uncertainties, which are of key importance in any kind of scientific experiment.

## Supporting information

**S1 Table. Periodic decomposition: Relative deviation of the error propagation simulated by Monte-Carlo and the basic image error $\Delta I$ for $N = 10^5$ iterations.**
(PDF)

**S2 Table. Power spectrum: Relative deviation between calculated and Monte-Carlo simulated uncertainty for $N = 10^5$ iterations.**
(PDF)

**S1 Fig. Sensitivity analysis of the central cut-off radius.** (A) Squared difference between calculated angular orientation distribution and sampled fiber angle distribution on $N = 10^4$ images (Monte-Carlo generated). The same dataset was also used for evaluation parameter optimization. Unfiltered power spectra are used to isolate the effect of the cut-off radius. (B-D) Evaluation example for cut-off radii (0,3,6). The artifact dramatically reduces from a 0 pixel cut-off to a radial cut-off of 3 pixels. Cut-off radii above 3 pixels barely influence the angular distribution. In order to save angular information, we choose a cut-off radius of 3 pixels.
(EPS)

**S2 Fig. Optimal evaluation parameters vs. fiber properties and image noise.** Plotted are median values and errorbars represent the interquartile range (25, 75). (A,C,E,G) represent plots of $\delta_{cut}$ vs. dispersion $k$, noise factor NF, aspect ratio (AR) and width. (B,D,F,H) are plots of the weighting factor $\alpha$ vs. dispersion $k$, noise factor NF, aspect ratio (AR) and width.
(EPS)

**S3 Fig. Error of the dispersion parameter $k$ and $b$ of the von-Mises fit and the sigmoid fit of the AF method for fibrous Monte-Carlo images (width = 5, AR = 30).** The von-Mises fit was applied to the angular orientation distribution as calculated by the AF method, whereas the sigmoid fit was applied to the respective cumulative orientation distribution.
(EPS)

**S1 Dataset. The .csv file provides the reference parameter as well as the parameter calculated by the AF method for each image.**
(CSV)

## Author Contributions

**Conceptualization:** Maximilian Witte, Michael Rübhausen, Frank Fischer.

**Data curation:** Frank Fischer.

**Software:** Maximilian Witte.

**Supervision:** Sören Jaspers, Horst Wenck, Michael Rübhausen, Frank Fischer.

**Validation:** Maximilian Witte.

**Visualization:** Maximilian Witte.

**Writing – original draft:** Maximilian Witte.

**Writing – review & editing:** Maximilian Witte, Michael Rübhausen, Frank Fischer.

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
