## [Decision Letter · Decision Letter 0]

11 Nov 2019

PONE-D-19-26997

Noise Reduction and Quantification of Fiber Orientations in Greyscale Images

PLOS ONE

Dear Mr. Witte,

Thank you for submitting your manuscript to PLOS ONE. After careful consideration, we feel that it has merit but does not fully meet PLOS ONE’s publication criteria as it currently stands. Therefore, we invite you to submit a revised version of the manuscript that addresses the points raised during the review process.

This revised manuscript does a good job of addressing the comments from the previous submission. There is one significant change suggested by reviewer 2 regarding the comparison with the band-pass method. If this comment is addressed, I believe that the manuscript will be suitable for publication.

We would appreciate receiving your revised manuscript by Dec 26 2019 11:59PM. To enhance the reproducibility of your results, we recommend that if applicable you deposit your laboratory protocols in protocols.io, where a protocol can be assigned its own identifier (DOI) such that it can be cited independently in the future. For instructions see: http://journals.plos.org/plosone/s/submission-guidelines#loc-laboratory-protocols

We look forward to receiving your revised manuscript.

Kind regards,

William Speier, PhD

Academic Editor

PLOS ONE

Journal Requirements:

2. We note that Figures in your submission contain images which may be copyrighted.

a. You may seek permission from the original copyright holder of Figures to publish the content specifically under the CC BY 4.0 license.

3. Please amend your Data availability statement and Methods section to clearly indicate the origin of all images used for testing the method. Please also clarify if any images were obtained specifically for this study, and if so, whether these have been made available to other researchers.

4. Your ethics statement must appear in the Methods section of your manuscript.

If your ethics statement is written in any section besides the Methods, please move it to the Methods section and delete it from any other section.

Please also ensure that your ethics statement is included in your manuscript, as the ethics section of your online submission will not be published alongside your manuscript.

5. Please provide the specific IRB approval number.

Reviewers' comments:

Reviewer's Responses to Questions

**Comments to the Author**

1. Is the manuscript technically sound, and do the data support the conclusions?

Reviewer #1: Yes

Reviewer #2: Partly

2. Has the statistical analysis been performed appropriately and rigorously? 

Reviewer #1: Yes

Reviewer #2: Yes

3. Have the authors made all data underlying the findings in their manuscript fully available?

Reviewer #1: Yes

Reviewer #2: No

4. Is the manuscript presented in an intelligible fashion and written in standard English?

Reviewer #1: Yes

Reviewer #2: Yes

5. Review Comments to the Author

Reviewer #1: Presented manuscript provides new approach for fiber orientation analysis. The power spectrum filtering based on relative uncertainity is very nice idea. I believe this method will help researchers in the field of image processing improve their algorithms for fiber orientation estimation. All my previous concerns were adequately addressed so I suggest to accept this paper.

Reviewer #2: The authors did a nice job addressing many of the comments from the original review. While I do feel that the quality of the paper has been considerably strengthened, I feel there is a critical deficiency that should be corrected prior to publication:

A direct comparison is now made with a band-pass method (Morrill et al., FiberFit). This is a very helpful analysis; however, the results are difficult to interpret since the adaptive filtering method (AF) uses preprocessed images while it appears that the band-pass method uses the raw images. More specifically, the AF method uses periodic decomposition of the raw image to remove cross-like artifacts in the Fourier domain (Fig. 2), but it appears that these edge artifacts were not removed from the images used for the band-pass analysis. In the paper by Morrill et al., the test images were generated without high intensity pixels near the image boundary to avoid cross-like artifacts in the Fourier domain (similar to a window function). This difference in preprocessing may contribute to the large fiber dispersion error calculated in this paper for the band-pass method (34.3%) relative to the fiber dispersion error calculated for the band-pass method in Morrill et al. (7.4%). Since the importance of edge artifact removal is well known (Moisan 2011) and is not a novel contribution of this paper, preprocessed images that remove this edge artifact (via periodic decomposition) should be used in the analysis of both the AF and band-pass methods. This would allow a more direct interpretation of how the novel contributions of this study effect error in measuring fiber dispersion and mean fiber orientation, relative to the band-pass method.

Other comments:

- In the original submission, figure 8 was included to show the advantage of using a Sigmoid function vs Von-Mises function. It appears that this was removed, and there is no longer any quantitative or qualitative comparison with a Von-Mises function. Therefore, the advantage of a Sigmoid function is no longer apparent w.r.t. a conventional Von-Mises approach. A figure showing the difference in error using these two different methods should be returned to the paper (could be a supplemental figure).

- For calculating fiber dispersion error for the AF method, please make it clear how you calculated percent error between the parameter (k) input into the von-Mises function to generate the test images, and the parameter (b) that was obtained by fitting the sigmoid function.

- Please include the magnification of the SHG images, and include a scale bar on the images in Fig. 8.

- For transparency, all generated images used to compare the AF method and the band-pass method should be made publicly available via a public repository.

6. PLOS authors have the option to publish the peer review history of their article (what does this mean?). If published, this will include your full peer review and any attached files.

Reviewer #1: No

Reviewer #2: No

---

## [Author Response · Author response to Decision Letter 0]

10 Dec 2019

Major comment (summarized):

Use preprocessed (periodic plus smooth decomposition) images for the comparison of the band-pass method with the AF method.

Response:

We fully agree with your point. We noticed, that our generated MC images rarely suffer from any cross-like artifacts. However, we re-calculated the errors and levels of significance using the smooth component of the decomposition for the band-pass method. The global error of the dispersion coefficient k sligthly decreased from (34.3±26.7)% to (33.9±26.5)%.

Other comments:

- In the original submission, figure 8 was included to show the advantage of using a Sigmoid function vs Von-Mises function. It appears that this was removed, and there is no longer any quantitative or qualitative comparison with a Von-Mises function. Therefore, the advantage of a Sigmoid function is no longer apparent w.r.t. a conventional Von-Mises approach. A figure showing the difference in error using these two different methods should be returned to the paper (could be a supplemental figure).

Response:

We agree on that. We adjusted the figure to the current dataset and added it to our supplement (S3_Fig). In addition, we refer to the supplemental figure in the discussion of the revised manuscript. 

- For calculating fiber dispersion error for the AF method, please make it clear how you calculated percent error between the parameter (k) input into the von-Mises function to generate the test images, and the parameter (b) that was obtained by fitting the sigmoid function.

Response:

As stated in our manuscript (lines 228-232) we fitted both, the frequency distribution and the cumulative frequency distribution of the sampled fiber angles to obtain reference parameter b and k. Using this approach, we additionally account for statistical sampling errors. If we use the predefined value of k as reference value we get an error of (35.0±25.6)% as global error for the band-pass method (preprocessed images).

- Please include the magnification of the SHG images, and include a scale bar on the images in Fig. 8.

- For transparency, all generated images used to compare the AF method and the band-pass method should be made publicly available via a public repository.

Response:

We also included a scale bar on the images and uploaded the data to a figshare repository.

---

## [Decision Letter · Decision Letter 1]

23 Dec 2019

Noise Reduction and Quantification of Fiber Orientations in Greyscale Images

PONE-D-19-26997R1

Dear Dr. Witte,

We are pleased to inform you that your manuscript has been judged scientifically suitable for publication and will be formally accepted for publication once it complies with all outstanding technical requirements.

With kind regards,

William Speier, PhD

Academic Editor

PLOS ONE

Additional Editor Comments (optional):

Reviewers' comments:

Reviewer's Responses to Questions

**Comments to the Author**

1. If the authors have adequately addressed your comments raised in a previous round of review and you feel that this manuscript is now acceptable for publication, you may indicate that here to bypass the “Comments to the Author” section, enter your conflict of interest statement in the “Confidential to Editor” section, and submit your "Accept" recommendation.

Reviewer #2: All comments have been addressed

2. Is the manuscript technically sound, and do the data support the conclusions?

Reviewer #2: Yes

3. Has the statistical analysis been performed appropriately and rigorously? 

Reviewer #2: Yes

4. Have the authors made all data underlying the findings in their manuscript fully available?

Reviewer #2: Yes

5. Is the manuscript presented in an intelligible fashion and written in standard English?

Reviewer #2: Yes

6. Review Comments to the Author

Reviewer #2: (No Response)

7. PLOS authors have the option to publish the peer review history of their article (what does this mean?). If published, this will include your full peer review and any attached files.

Reviewer #2: No

---

## [Editor Report · Acceptance letter]

7 Jan 2020

PONE-D-19-26997R1 

Noise Reduction and Quantification of Fiber Orientations in Greyscale Images 

Dear Dr. Witte:

I am pleased to inform you that your manuscript has been deemed suitable for publication in PLOS ONE. Congratulations! Your manuscript is now with our production department. 

With kind regards,

on behalf of

Dr. William Speier 

Academic Editor

PLOS ONE